# Genome-Wide Identification and Expression Analysis of TONNEAU1 Recruited Motif (TRM) Gene Family in Tomato

**DOI:** 10.3390/ijms26083676

**Published:** 2025-04-13

**Authors:** Xinyi Jia, Qingjun Fu, Guohao Yang, Xinyi Zhang, Xiaoluan Zhao, Yingying Nie, Chunying Feng, Jiayi Gao, Siyu Zhang, Manman Li, Haoran Wang, Xue Gong, Ying Han, Jingfu Li, Xiangyang Xu, Jingbin Jiang, Huanhuan Yang

**Affiliations:** Key Laboratory of Biology and Genetic Improvement of Horticultural Crops (Northeast Region), Ministry of Agriculture and Rural Affairs, College of Horticulture and Landscape Architecture, Northeast Agricultural University, Harbin 150030, China

**Keywords:** *Solanum lycopersicum*, gene family analysis, TONNEAU 1 recruited motif, gene expression, fruit shape

## Abstract

The TONNEAU1 Recruited Motif (TRM) gene family is integral to the growth and development of various plants, playing a particularly critical role in regulating the shape of plant organs. While the functions of the TRM gene family in other plant species have been documented, knowledge regarding the members of the tomato (*Solanum lycopersicum*). SlTRM gene family remains limited, and their specific roles are not yet well understood. In this study, we identified and analyzed 28 members of the SlTRM gene family in tomato using bioinformatics approaches based on the latest whole genome data. Our analysis included the examination of protein structures, physicochemical properties, collinearity analysis, gene structures, conserved motifs, and promoter cis-acting elements of the SlTRM gene family members. The phylogenetic analysis indicated that both tomato and *Arabidopsis thaliana* are categorized into five distinct subfamilies. Furthermore, we conducted a three-dimensional structure prediction of 28 SlTRM genes for the first time, utilizing AlphaFold3, a deep learning architecture developed by DeepMind. Subsequently, we analyzed public transcriptome data to assess the tissue specificity of these 28 genes. Additionally, we examined the expression of SlTRM genes using RNA-seq and qRT-PCR techniques. Our analysis revealed that *SlTRM25* was significantly differentially expressed, leading us to hypothesize that it may be involved in the development of lateral branches in tomatoes. Finally, we predicted the regulatory interaction network of *SlTRM25* and identified that it interacts with genes such as *SlFAF3/4b*, *SlCSR-like1*, *SlCSR-like2*, and *SlTRM19*. This study serves as a reference for the investigation of the tomato TRM gene family members and introduces a novel perspective on the role of this gene family in the formation of lateral branches in tomatoes, offering both theoretical and practical significance.

## 1. Introduction

The tomato (*Solanum lycopersicum*) is among the most widely cultivated vegetable crops globally, and its fruit possesses significant nutritional and economic value [1]. The shape of the tomato fruit is a crucial agronomic trait, exhibiting considerable diversity from wild tomatoes to domesticated and improved cultivated varieties [2]. Interestingly, wild tomato fruits are predominantly round, whereas cultivated tomato fruits exhibit a variety of shapes, including flat, round, elongated, pear-shaped, plum-shaped, and bell pepper-shaped forms [3,4,5]. These diverse shapes influence their commodity use and value; for example, slender and lumpy fruit shapes are ideal for processing into tomato sauce, cherry-shaped fruits are suitable for consumption, and round fruits are preferred for slicing [6]. In today’s rapidly advancing agricultural landscape, a well-shaped fruit that is suitable for mechanical harvesting can significantly reduce both labor costs and time [7]. Recent studies have elucidated the regulatory mechanisms underlying tomato fruit shape at multiple levels, including QTL gene regulation (*GLOBE*, *HAPT3*), hormonal regulation (*ARF7*, *NCED1*), and transcriptional regulation (*KUN*, *mi319*) [5,8,9,10,11,12,13,14]. Various gene family members within the genome are interspersed, creating a complex network system [15]. Therefore, identifying and further elucidating the gene family members that regulate the shape of tomato fruits is crucial for manipulating the morphology of agricultural products, facilitating breeding efforts, and enhancing the value of commodities.

The motif protein recruited by TONNEAU1 is named for its interaction with TONNEAU1 (TON1). This protein shares a conserved motif with human centrosomal proteins and is essential for microtubule organization in *Arabidopsis*, playing a crucial role in plant growth and morphogenesis [6,16]. For instance, mutants of *Arabidopsis thaliana* lacking *AtTRM5* exhibit a phenotype characterized by slow leaf growth and delayed flowering. Additionally, the knockout of *AtTRM6/AtTRM61* results in seed dysplasia and embryonic lethality, significantly impairing the growth and development of *Arabidopsis thaliana* [17]. In contrast, deletion mutants of *Arabidopsis AtTRM7* and *AtTRM8* exhibited impaired early band formation during the premitotic stage; however, they did not affect the formation of interphase microtubules [18]. Microtubules, as a crucial component of the plant cytoskeleton, play a significant role in plant growth and development, cell morphology, and immune responses to various stresses. Members of the TRM gene family play a crucial role in the formation and structural stability of microtubules [19,20,21,22,23]. Additionally, TRM family members interact with TON1 and protein phosphatase 2A (PP2A) to form a TPP (TON1-TRM-PP2A) protein complex through their M2 and M3 motifs, respectively [24,25,26]. This complex regulates the formation of PPB in the proplanar division of tissue cells, which subsequently influences the microtubule organization, altering the cell division pattern and resulting in the production of fruit organs with varying shapes [24,27,28]. In *Arabidopsis thaliana*, the *AtTRM1* gene utilizes its C-terminal M2 motif to interact with TON1 in vivo, thereby targeting TON1 to cortical microtubules [28]. The wheat *GW7* gene promotes shape elongation by enhancing cell proliferation at the distal end [29]. Additionally, TRM interacts with ovarian family protein (OFP) genes via its M8 motif to regulate cell division and organ growth, influencing the interactions between the cytoplasm and microtubules [30,31,32]. Although OFP*s* have been identified as transcriptional repressors in previous studies, recent data indicate that this family of genes also plays a role in regulating organ shape by maintaining a dynamic balance through their regulatory localization [33,34]. The OFP-TRM protein complex may also collaborate with other shape-regulating proteins, such as IQDs, to establish stable organ shapes by modulating cytoskeletal activity in response to external stimuli and hormonal signals [35]. Consequently, it is crucial to investigate the regulatory mechanisms through which TRM family members contribute to the regulation of fruit shape as essential components.

In various horticultural crops, fruit shape is a primary focus for researchers. Tomatoes serve as an excellent model crop for studying the growth of fleshy fruits and the variation in organ shape, with numerous genes identified that control these traits. To date, many genes regulating tomato fruit shape have been reported. For instance, the regulatory mechanisms of genes such as *SUN*, *OVATE*, *SlOFP20* (*SOV1*), *LC*, *FAS*, and *fs8.1* have been elucidated [1,36,37]. Plant TRM gene families are known to play significant roles in the regulation of fruit shape. Researchers have discovered that in cucumber, the homologous gene *CsTRM5*, which corresponds to the tomato gene *SlOFP20*, is crucial for the elongation of cucumber fruit [38]. In rice, the TRM homologous gene *OsGL7/SLG7* forms a protein complex with TON1 and PP2A through its M2 and M3 domains, respectively [39,40]. This interaction regulates cell length and width, which subsequently influences grain size and quality. Among the members of the tomato TRM gene family, *SlTRM5* has been demonstrated to promote the elongation of tomato fruit by regulating cell division, whereas *SlTRM3* and *SlTRM4* have a lesser impact on tomato fruit development [30,38]. Interestingly, the phenotypes of the *SlTRM3/4* and *SlTRM5* diploids in the background of tomato *LA1589* tended to resemble that of the *SlTRM5* single mutant, indicating that *SlTRM3/4* and *SlTRM5* exert additive effects in regulating fruit shape [6]. Additionally, tomato *SlTRM17/20a* and *SlTRM19* co-regulate elongation, whereas *SlTRM26a* has a limited impact on fruit shape [6]. In addition to the synergistic effects observed among members of the TRM gene family, *SlTRM5* and *SlTRM19* exhibit antagonistic effects in the regulation of fruit shape. Emerging evidence demonstrates that TRM genes modulate tomato fruit shape by regulating cell proliferation patterns along growth axes. Notably, *SlTRM3/4* and *SlTRM5* mutations restore the elongated phenotype of *OVATE/SlOFP20* mutants to round fruits. Evolutionary conserved TRM members (*Arabidopsis TRM1/2*, *SlTRM5*, and cereal *GW7*) promote elongation by enhancing proximal–distal cell proliferation while suppressing medio-lateral division. These findings suggest TRM-mediated microtubule dynamics coordinate cellular patterning during fruit morphogenesis. Despite the established roles of *SUN* and *OVATE* in fruit shape determination, TRM family functions remain poorly characterized, with current studies limited to leaf development and fruit-specific mechanisms awaiting investigation [38]. Therefore, it is worthwhile to investigate whether other members of the SlTRM gene family contribute to the regulation of tomato fruit morphology and are involved in the determination of other tomato shapes.

While numerous members of the tomato SlTRM gene family have been identified and studied, a systematic bioinformatics analysis has yet to be conducted on those specifically related to tomato fruit formation. In this study, we identified 28 SlTRM genes utilizing the latest genomic data from tomatoes and analyzed their physical and chemical properties, compliance, phylogenetic relationships, and cis-acting elements. Additionally, we predicted the three-dimensional structures of SlTRM family members for the first time using AlphaFold3. We also examined the tissue-specific expression of these SlTRM family members. Through RNA-Seq and qRT-PCR analyses, we discovered that *SlTRM25* may play a role in regulating the development of tomato lateral clades. Furthermore, we predicted the interaction network of *SlTRM25* and explored its role in plant growth and development. This study contributes to a deeper understanding of the regulatory mechanisms of SlTRM gene family members concerning tomato fruit shape and collateral development.

## 2. Results

### 2.1. Identification and Characterization of TRM Genes in Tomato

Gene family members were initially screened in the tomato genome using BLAST 2.1 and hmmsearch 3.0. Following this, SMART and NCBI CD-search were employed in conjunction with SGN gene annotation to verify the candidate genes, ultimately leading to the identification of 28 SlTRM genes. The distribution of these genes across tomato chromosomes is illustrated in Figure 1. The SlTRM gene is evenly distributed among 11 chromosomes, primarily located in the terminal regions of these chromosomes. Notably, chromosome 9 harbors the highest number of SlTRM genes, with a total of five, while chromosomes 4, 5, and 12 each contain at least one gene.

The analysis of physical and chemical properties revealed that the number of amino acids in the SlTRM family members ranges from 220 (*SlTRM3*) to 3279 (*SlTRM3/4*), with their molecular weights varying from 8.386 kDa to 121.657 kDa. Further details regarding the theoretical isoelectric points, instability indices, hydrophilicity, and hydrophobicity of the SlTRM family members can be found in the Appendix A. Interestingly, all members of the SlTRM family are hydrophilic proteins. Additionally, subcellular localization prediction results revealed that most genes are localized in the nucleus, while *SlTRM22* and *SlTRM1/2/3/4/5* are localized in the chloroplast, and *SlTRM17/20a* and *SlTRM17/20b* are localized in peroxisomes. This distribution indicates the functional diversity among the members of the SlTRM gene family.

### 2.2. Phylogenetic Relationship Analysis of TRM Genes in Tomato

To investigate the phylogenetic relationships of the tomato SlTRM gene, we selected 34 *Arabidopsis* AtTRM protein sequences and 28 tomato SlTRM protein sequences for the construction of phylogenetic trees. The results indicated that the phylogenetic trees were categorized into five subfamilies (I, II, III, IV, V), with both tomato and *Arabidopsis* TRM proteins distributed across each subfamily (Figure 2). Among the groups, clade I and II members comprise up to seventeen and sixteen proteins, respectively, while subgroup IV contains the fewest genes, comprising only five proteins (AtTRM30, SlTRM30, AtTRM34, SlTRM30/34a, and SlTRM30/34b). In summary, the tomato and SlTRM members of the AtTRM family in *Arabidopsis* are closely related and exhibit relatively conserved evolutionary relationships. Research in Arabidopsis has established distinct functions for TRM family members: *AtTRM5* modulates leaf morphogenesis through microtubule reorganization, *AtTRM7* regulates cell division patterns, and *AtTRM21* participates in flavonoid biosynthesis. Based on these evolutionarily conserved mechanisms, we hypothesize that their tomato orthologs may maintain analogous functions in growth regulation and metabolic pathways. These findings provide critical insights into TRM-mediated developmental processes in tomato and establish a foundation for the future functional characterization of this gene family [18,41,42].

### 2.3. Conserved Motif and Gene Structure Analysis of TRM Family Members in Tomato

MEGA11 was employed to construct phylogenetic trees for 28 members of the SlTRM gene family, and their gene structures and conserved motifs were subsequently analyzed. The phylogenetic analysis indicated that the SlTRM genes were categorized into multiple branches, suggesting potential evolutionary divergences among the SlTRM genes (Figure 3A). Subsequently, we analyzed the conserved motifs and gene structures of the SlTRM family members. Our findings revealed that most SlTRM genes exhibited structural similarities (Figure 3C), with *SlTRM3* and *SlTRM6* containing only one coding sequence (CDS). The tomato TRM family members (*SlTRM5*, *SlTRM19*, and *SlTRM17/20a*) exhibit multiple splicing variants and regulate fruit morphogenesis through microtubule-dependent cytoskeleton reorganization. The comparative structural analysis indicates that *SlTRM3* and *SlTRM6* have simpler structures, suggesting they may function through distinct spatiotemporal mechanisms in tissue-specific developmental regulation. The conservation motif analysis identified a total of 10 conserved motifs across all SlTRM genes, with motifs 1, 3, and 8 being present in the majority of these genes (Figure 3B). This indicates a high level of conservation of these motifs within the gene family. The conserved motifs were further visualized using WEBLOGO, revealing that motif 8 was the shortest, while motifs 9 and 10 were the longest (Figure 3D). In conclusion, these motifs play a crucial role in maintaining the structural integrity and functional activity of SlTRM genes.

### 2.4. Prediction of the Three-Dimensional Structures of SlTRM Proteins

The protein structure encoded by a gene is essential for its function. AlphaFold3 is an artificial intelligence model designed for high-precision biomolecular structure prediction, utilizing deep learning technology. We employed the AlphaFold3 platform to predict the three-dimensional structures of 28 members of the SlTRM gene family in tomatoes (Figure 4). Through the detection and evaluation of online software, the three-dimensional structure of the SlTRM gene family is believed to be accurate. Notably, the three-dimensional structures of the *SlTRM3* and *SlTRM6* proteins are relatively simple when compared to those of other genes. The canonical *SlTRM5* displays a predominantly disordered conformation (82.75% random coil) with a minimal α-helical content (16.38%), contrasting sharply with the well-ordered structures of *SlTRM3* (66.20% α-helix; 33.80% random coil) and *SlTRM6* (64.60% α-helix; 35.96% random coil). This pronounced structural dichotomy strongly suggests functional divergence: *SlTRM5’s* intrinsic disorder likely facilitates transient protein–protein interactions in signaling complexes, while *SlTRM3/6’s* stable helical architectures may serve structural roles in cytoskeletal organization. We hypothesize that these two genes contain only the folded domain of the core, which establishes a foundation for future investigations into the functions of the members of the SlTRM gene family.

### 2.5. Prediction of Cis-Acting Elements of SlTRM Family Members

Cis-acting elements, as regulatory targets of upstream genes, play a crucial role in the transcription, expression, and function of genes [43]. In this study, we selected 28 SlTRM gene transcription start checkpoints located within 2000 base pairs upstream as promoter sequences for prediction using the PlantCare website, identifying a total of 20 cis-acting elements (Figure 5A). These cis-acting elements are categorized into several types: developmental-related elements, environmental-stress-related elements, hormone-stress-responsive elements, light-responsive elements, promoter-related elements, and site-binding-related elements (Figure 5B). Among the members of the SlTRM gene family, light-responsive elements and hormone-responsive elements are the most prevalent. These elements may play a role in the transcriptional regulation of genes by serving as binding checkpoints. Specific details are provided in Appendix A.

### 2.6. Collinearity Analysis of TRM Family Members in Tomato

Gene duplication events play a crucial role in gene amplification and the expansion of gene families during genome amplification [44]. These genes are distributed across nearly all chromosomes; however, certain chromosomes, such as Chr01 and Chr09, exhibit higher densities of SlTRM genes, suggesting these regions may be significant for gene amplification or duplication. The analysis of intraspecific gene collinearity within the tomato SlTRM gene family revealed collinearity among six genes across twenty-eight SlTRM gene families, suggesting that these SlTRM genes may share similar functions (Figure 6A). Additionally, the gray lines illustrate the collinearity among the remaining genes in the tomato genome, and these collinearity blocks demonstrate the conservation of genes throughout evolution. To further investigate the evolutionary relationships among TRM members, we conducted an interspecies collinearity analysis involving *Solanum lycopersicum*, *Arabidopsis thaliana*, *Solanum tuberosum*, and *Capsicum annuum* (Figure 6B–D). In this analysis, numerous genes exhibiting collinearity with tomato TRM genes were identified across these species. Notably, *Arabidopsis thaliana* TRM genes displayed the highest number of collinear genes with those in tomato, while *Capsicum annuum* TRM genes exhibited the fewest. These findings indicate a closer evolutionary conservation between *Arabidopsis thaliana* and *Solanum lycopersicum*, providing a valuable reference for the analysis of genetic relationships and gene functions among these species.

### 2.7. Expression of SlTRM Genes in Different Tissues and Organs of Tomato

We observed the expression of 28 genes across different tissues, with the majority being expressed in all examined tissues (Figure 7). Notably, *SlTRM13/14/15/33a*, *SlTRM17/20a*, *SlTRM17/20b*, and *SlTRM22* exhibited high expression levels in fruit, whereas *SlTRM16/32b*, *SlTRM16/32c*, and *SlTRM26b* demonstrated very low expression levels across all tissues. Additionally, *SlTRM6* was not expressed in any tissue. Interestingly, most genes showed higher expression levels in fruit tissues compared to other tissues, suggesting a potential role in regulating fruit development (Appendix A).

### 2.8. Expression Patterns of SlTRM Genes Revealed by Lateral Development Transcriptome Analysis and qRT-PCR Analysis

In this study, we analyzed the expression patterns of 28 genes within the lateral development transcriptome. Our findings revealed that 17 genes exhibited elevated expression levels in the overexpressed plants compared to the control, whereas 11 genes demonstrated reduced expression levels in the overexpressed plants relative to the control (Figure 8A). Detailed information regarding specific expression volumes (FPKM) is available in Appendix A. To validate the transcriptome data, we selected four genes for qRT-PCR analysis (Figure 8C–F). As illustrated in the figure, the expression trends of these genes were consistent with those observed in the transcriptome data. Our findings indicate that *SlTRM25* is significantly differentially expressed when compared to other TRM gene families. We hypothesize that *SlTRM25* may play a regulatory role in the development of lateral branches in tomato. Simultaneously, we conducted a Gene Ontology (GO) enrichment analysis of 28 SlTRM genes in tomato using an online platform (Figure 8B). The results indicated that all genes are associated with cell growth and development, with SlTRM genes primarily enriched in the monopolar cell growth and cell morphogenesis pathways. This finding suggests that SlTRM genes may play a role in the development of tomato collateral, providing a basis for future studies on downstream genes involved in tomato collateral development.

### 2.9. Analysis of SlTRM25 Gene Expression Network

We conducted an interaction network prediction for *SlTRM25* using the STRING platform, which revealed that ten genes interact with *SlTRM25* (Figure 9). In addition to four unknown proteins, these include the delayed flowering gene *SlFAF3/4b*, the tomato fruit development genes *SlCSR-like1* and *SlCSR-like2*, the heat-stress-related gene *Solyc02g085030*, and the negative regulatory gene *SlTRM19*, which controls fruit elongation. Based on the preliminary data from our research group, we discovered that *SlTRM25* exhibits a similar expression pattern to known regulators of lateral branch development (*BL*, *BRC1b*, and *PIN4*), showing upregulation in overexpression lines [45]. These genes have been shown to exert varying degrees of influence on tomato growth and development. In summary, we propose that *SlTRM25* may play a role in regulating the development of lateral branches and the shape of tomato fruit.

## 3. Discussion

The branch structure and fruit organs of plants are critical agronomic characteristics for horticultural crops [46]. Currently, enhancing plant structure and regulating fruit organ development are essential for ensuring stable crop yields and achieving breakthroughs in production. Optimal plant configuration and well-formed fruit organs are necessary to facilitate mechanized harvesting, thereby improving work efficiency and reducing labor costs [47]. In the process of crop domestication, enhancing plant structure and the shape of fruit organs through molecular design is a crucial strategy for improving crop yield [48,49,50]. For instance, wild ruminant grass exhibits structural multi-branching, whereas corn with a mutation in the *TB1* (*Ruminant Branch 1*) gene displays a lack of branching [45,51]. During the evolution from wild tomato to cultivated tomato, a significant diversity in fruit shape was observed. Specific genes, such as *IAA17*, *CLV3/FAS*, and *CLV1/FAB*, have been identified as regulators of fruit shape. Therefore, a further elucidation of the specific molecular mechanisms that govern plant structure and fruit shape is crucial for understanding plant domestication and enhancing crop genetics [52,53].

TONNEAU1 Recruited Motif (TRM) proteins are crucial for plant growth and morphogenesis. Members of the TRM family not only interact with TON1 and OVATE, but also bind to cortical microtubules, thereby regulating plant growth and development [28,44]. While mutations in the TRM gene family have been associated with alterations in tomato fruit shape, a systematic analysis of the TRM gene family in tomato has yet to be conducted to explore the common characteristics of its members. In this study, we identified 28 SlTRM genes within the tomato genome using bioinformatics techniques. These genes were found to be distributed across 11 chromosomes, with the majority located at the chromosomal termini. Furthermore, the results of subcellular localization predictions indicated that SlTRM genes are present not only in the nucleus but also in the peroxisome and chloroplast, suggesting that they may serve diverse functions. To investigate the evolutionary relationship of tomato SlTRM genes, we selected TRM genes from both Arabidopsis and tomato for phylogenetic analysis. The results indicated that the TRM genes from Arabidopsis and tomato can be classified into five distinct subgroups. Throughout plant evolution, certain conserved motifs have been retained for their functional significance; thus, genes exhibiting similar structures and motifs frequently share analogous functions [54,55]. We examined the gene structures and conserved motifs of the SlTRM gene family members. Our findings indicate that conserved core motifs are present in the majority of the genes, while some genes contain only one or two specific motifs. This suggests that these genes have undergone conservation and evolution over time. Additionally, the AlphaFold3 motif deep learning framework, developed by Google and DeepMind, facilitates high-precision biomolecular structure prediction [56]. The algorithm can predict the three-dimensional structure of proteins and the interactions between molecules based on their corresponding protein and nucleotide sequences. We utilize this algorithm to accurately predict the three-dimensional structures of 28 SlTRM proteins, hypothesizing that the manner in which these proteins curl and fold is closely related to their structural functions. Cis-acting elements within the promoter region do not encode proteins; however, they play a crucial role in regulating gene expression in conjunction with trans-acting factors. For instance, mutations in various cis-acting elements of the promoter for the *WUSCHEL HOMEOBOX9* (*WOX9*) gene resulted in distinct phenotypes in both *Solanaceae* and *non-Solanaceae* plants, underscoring the significant role that these cis-regulatory elements play in crop improvement [57,58]. In this study, we identified 20 cis-acting elements across 28 promoter regions of the SlTRM gene, which encompass light response, environmental stress, and other development-related elements. These cis-acting elements enable SlTRM to perform diverse functions in plant growth and development.

Gene replication events play a crucial role in plant evolution. Following domestication and selection, gene family members emerge as a group of genes with similar protein structures, a process facilitated by whole genome duplication (WGD) or other genome replication events [59,60]. These gene replication events can be categorized into whole genome replication and single gene replication [61,62]. Genome-wide replication is often lost or silenced over time, whereas single-gene replication tends to be preserved due to the diversity of replication methods. An intraspecies collinearity analysis of tomato SlTRM genes revealed that only six genes in tomato were collinearly related, indicating a lower degree of gene replication among SlTRM gene family members. Concurrently, to explore the homology of tomato with other plant TRM gene family members, collinearity relationships between tomato and species such as *Arabidopsis thaliana*, *Solanum tuberosum*, and *Capsicum annuum* were constructed. Tomato and *Arabidopsis* exhibit the highest number of collinear genes, whereas pepper displays the least, suggesting a degree of conservation among genes across different species. The tissue-specific expression of genes is crucial for understanding their functions. Consequently, we investigated the tissue expression of 28 SlTRM genes using RNA-seq data. Notably, some of these genes, such as *SlTRM3*, *SlTRM4*, and *SlTRM5*, are relatively highly expressed in leaves and have been implicated in the regulation of leaf shape [38]. Furthermore, a majority of the genes show high expression levels in fruit, including *SlTRM19* and *SlTRM17/20a*, which have been confirmed to be associated with fruit shape. Additionally, Gene Ontology (GO) enrichment analysis revealed that most of the genes are linked to cell growth and development, reinforcing the conclusion that SlTRM genes play a role in regulating fruit shape [63,64].

We hypothesized that members of the SlTRM gene family play a role in regulating the development of lateral branches in tomatoes. All identified members of the SlTRM gene family in the transcriptome exhibited differential expression, with some genes being upregulated and others downregulated. This pattern suggests that the members of the SlTRM gene family may serve distinct functions during the development of lateral branching. Four differentially expressed genes were selected for quantitative reverse transcription polymerase chain reaction (qRT-PCR) analysis, and the results were consistent with the trends observed in the transcriptome data. Among these, *SlTRM25*, identified as a significantly differentially expressed gene, may be involved in the development of tomato lateral clades. Utilizing the STRING database, we further predicted the interaction network of *SlTRM25*. The findings revealed that *SlTRM25* interacts with genes associated with flowering, fruit development, and stress responses, suggesting that *SlTRM25* may have a significant role in plant growth and morphogenesis.

## 4. Materials and Methods

### 4.1. Tomato Growth and Treatment

Using cultivated Ailsa Craig as the experimental material, all tomato seeds were germinated under conditions of constant humidity and darkness at a temperature of 27 degrees Celsius. Subsequently, the seeds were planted in an artificial climate incubator. The program was set to maintain a temperature of 25 °C with light for 16 h at a relative humidity of 60%, followed by 20 °C in darkness for eight hours at a relative humidity of 50%. Throughout this period, tomato growth management practices were implemented to ensure that the plants maintained optimal growth conditions.

### 4.2. Identification and Physicochemical Properties Analysis of Gene Family Members

The latest genomic data for tomato were downloaded from the Solanaceae Genomics Network (SGN) website (https://solgenomics.net/ftp/tomato_genome/annotation/ITAG4.0_release/, accessed on 2 November 2024). Additionally, the genomic sequences, protein sequences, and annotation files (GFF3) for potato, pepper, and Arabidopsis thaliana were obtained from Ensembl (http://plants.ensembl.org/index.html, accessed on 2 November 2024) for the subsequent analysis of the SlTRM gene family members. Preliminary screening of the SlTRM family members was conducted using the protein sequences of the *Arabidopsis* AtTRM family members according to BLAST2.1 with an E-value threshold of 10^−5^. Subsequently, the conserved domains of SlTRM (PF14383 and PF14309) were downloaded from the InterPro website (https://www.ebi.ac.uk/interpro/, accessed on 2 November 2024) for further screening of SlTRM members using TBtools. Pseudogenes were then removed using the NCBI CD-search (https://www.ncbi.nlm.nih.gov/Structure/bwrpsb/bwrpsb.cgi/, accessed on 5 November 2024) and SMART (http://smart.embl-heidelberg.de/, accessed on 5 November 2024). Finally, SlTRM gene family members were identified in combination with SGN annotation features. The physicochemical parameters of the SlTRM proteins, including amino acid number, molecular weight, theoretical isoelectric point (pI), hydrophilic large mean (GRAVY), and instability index, were calculated using the ExPASy ProtParam tool (http://cello.life.nctu.edu.tw/, accessed on 8 November 2024). Additionally, PSORT (https://www.genscript.com/psort.html, accessed on 12 November 2024) was utilized to predict the subcellular localization of the proteins.

### 4.3. Chromosome Localization and Collinearity Analysis

The chromosome distribution and gene positions of the SlTRM gene family members were plotted using TBtools, following the methodology of Yang et al. [51]. Additionally, the collinearity among SlTRM gene family members, as well as between tomato, potato, Arabidopsis thaliana, and pepper species, was analyzed using TBtools 2.154 software [65].

### 4.4. Phylogenetic Analysis

A total of 34 AtTRM gene family members in *Arabidopsis thaliana* were identified by analyzing the SlTRM gene family members in tomato, in conjunction with data from the TAIR (https://www.arabidopsis.org/, accessed on 20 November 2024) website. Subsequently, 28 SlTRM gene members and 34 AtTRM members from Arabidopsis were sequenced using MEGA11. The sequences were pruned using TBtools for comparison, and a phylogenetic evolutionary tree was constructed using the Neighbor-Joining (NJ) method. The evolutionary tree was further enhanced through classification annotation using iTOL (https://itol.embl.de/, accessed on 20 November 2024).

### 4.5. Gene Structure and Conserved Domain Analysis

The structural location information of the SlTRM gene was obtained using the tomato genome annotation file (GFF3) and genome sequence extraction. The conserved motifs of the SlTRM gene family members were predicted using the MEME Suite (https://meme-suite.org/meme/tools/meme, accessed on 25 November 2024), which identified a total of 10 conserved motifs with the default threshold settings. The results were visualized using TBtools software.

### 4.6. Cis-Acting Element Analysis

The 2000 base pair (bp) promoter sequence located upstream of the transcriptional start site of the SlTRM gene was extracted. Subsequently, this promoter sequence was uploaded to the PlantCare website (http://bioinformatics.psb.ugent.be/webtools/plantcare/html/, accessed on 25 November 2024). Following this, the identified cis-acting elements were classified, organized, and statistically mapped based on the results.

### 4.7. Three-Dimensional Structure Prediction of Genes

The three-dimensional structure of the *SlTRM* protein was predicted using AlphaFold3, an advanced artificial intelligence model jointly developed by Google DeepMind and Isomorphic Labs (https://alphafoldserver.com/, accessed on 2 December 2024). The selection was based on the iPTM and PTM values derived from the predicted results. Additionally, the three-dimensional structure was visualized using UCSF ChimeraX software (https://www.cgl.ucsf.edu/chimerax/, accessed on 2 December 2024).

### 4.8. Gene Expression Pattern Analysis

Common transcriptome data from various tissues of the tomato variety Heinz 1706 were obtained from the Tomato Functional Genome Database (http://ted.bti.cornell.edu/cgi-bin/TFGD/digital/home.cgi, NCBI registration number SRA049915, accessed on 2 December 2024). These data were standardized to generate a heatmap based on mean FPKM values. The original transcriptome data for tomato lateral branches have been uploaded to the Gene Expression Omnibus (GEO) of NCBI (https://www.ncbi.nlm.nih.gov/, accessed on 2 December 2024) under accession number GSE164382.

### 4.9. RNA Extraction, qRT-PCR Verification, and Interaction Network Construction

Axillary buds of tomato plants were collected at 45 days post-germination, with three biological replicates obtained for each sample. Immediately following collection, the samples were stored in liquid nitrogen. The axillary bud tissues were then ground in a mortar, and total RNA was extracted from both wild-type AC and overexpressed plants using the TakaRa RNA extraction kit (Takara Bio, Beijing, China) in accordance with the manufacturer’s instructions. cDNA synthesis and gDNA removal were conducted using a reverse transcription kit from Vazyme (Vazyme, Nanjing, China). The resulting cDNA was assessed via gel electrophoresis and spectrophotometry, and subsequently stored at −80 °C as a template for future fluorescence quantification. Fluorescence quantitative primers for the selected genes were designed using Primer Premier 5.0 (refer to Appendix A), with the β-actin gene serving as the internal reference [66]. The real-time fluorescence quantitative PCR analysis system and the method for calculating relative expression amounts were adapted from Sun et al. [67]. The data were analyzed using GraphPad Prism 9, and statistical significance was assessed using a one-way analysis of variance, employing either non-parametric or mixed methods. The method for constructing the *SlTRM25* interaction network follows the approach outlined by Sun et al. [68].

## 5. Conclusions

Gene family analysis is a crucial method for gaining a deeper understanding of species characteristics and for analyzing homologous genes to explore the functions of plant evolution and morphological development. In this study, we identified 28 SlTRM genes for the first time using the latest tomato whole genome sequence and annotation file. We analyzed the physical and chemical characteristics, gene structure, conserved motifs, and cis-acting elements of the SlTRM genes, and constructed a phylogenetic tree depicting the evolutionary relationships of the TRM gene family in tomato and *Arabidopsis*. Interspecies and intraspecies collinearity analysis reveals the preservation of genomic replication during species evolution. Furthermore, we predicted the three-dimensional structures of the 28 SlTRM gene proteins using AlphaFold3. The tissue-specific expression analysis demonstrated that SlTRM genes exhibit differential expression across various tissues, suggesting that they perform distinct functions during plant growth and development. Transcriptome and qRT-PCR analyses indicated that the significantly differential expression of *SlTRM25* may play a role in the development of tomato lateral collaterals. STRING predictions suggest that *SlTRM25* may also be involved in the regulation of other processes in tomato growth and development. The results of this study provide a reference for further investigations of the tomato SlTRM gene family and offer new insights for the exploration of the function of the *SlTRM25* gene.

## Figures and Tables

**Figure 1 ijms-26-03676-f001:**
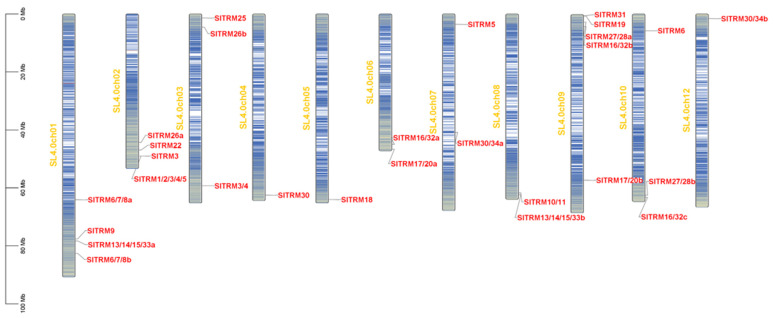
The position of SlTRM gene family members on chromosomes. A scale of 0 MB to 100 MB, representing the length of the chromosome. The yellow font indicates the number of chromosomes, arranged sequentially from left to right, while the line segments transitioning from blue to red on the chromosomes depict the density of genes.

**Figure 2 ijms-26-03676-f002:**
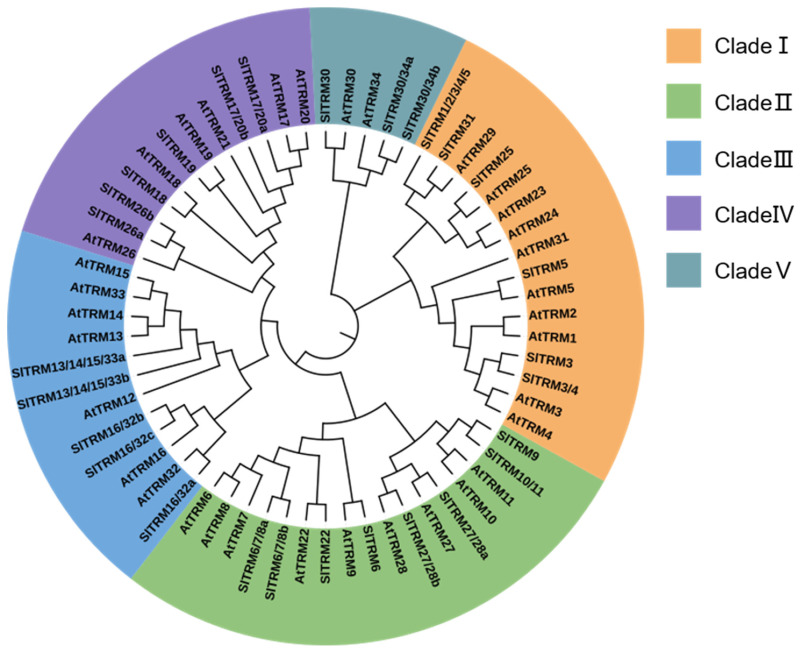
Phylogenetic tree of TRM family members from *Solanum lycopersicum* and *Arabidopsis thaliana*. The analysis reveals five distinct branches, each represented in a different color, corresponding to clades I, II, III, IV, and V.

**Figure 3 ijms-26-03676-f003:**
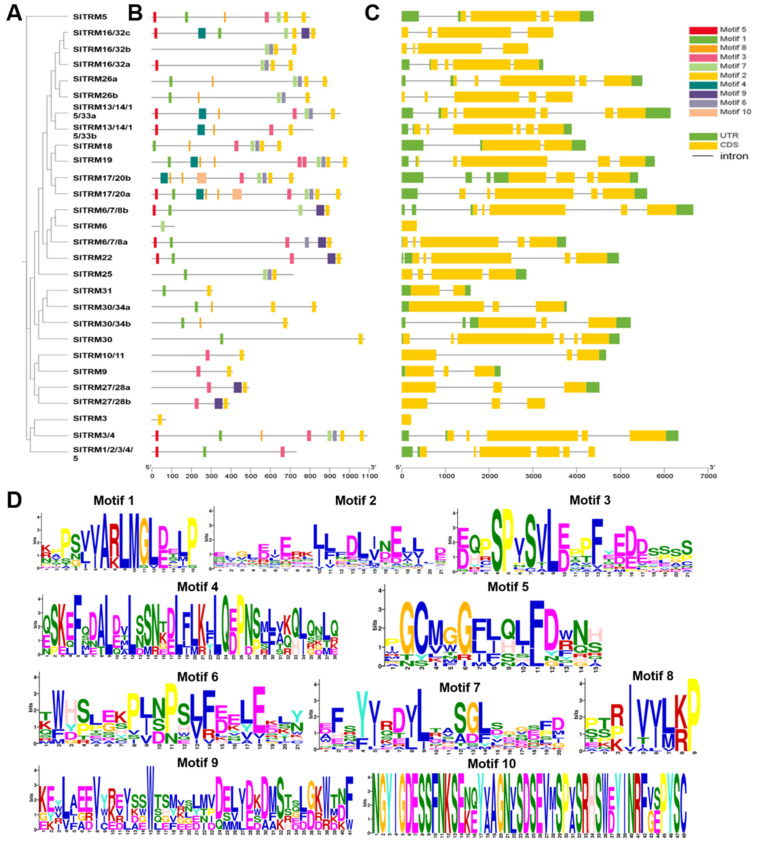
Gene structure and conserved motifs of SlTRM gene family members. (**A**) Phylogenetic tree depicting the relationships among SlTRM gene family members. (**B**) Distribution of conserved motifs within the SlTRM gene family, with different colors representing distinct conserved motifs. (**C**) Gene structure of SlTRM gene family members, where green indicates the UTR region, yellow denotes the CDS region, and horizontal lines represent the intron regions. (**D**) The amino acid sequences of the conserved motifs in SlTRM gene family members, with base size reflecting the frequency of occurrence.

**Figure 4 ijms-26-03676-f004:**
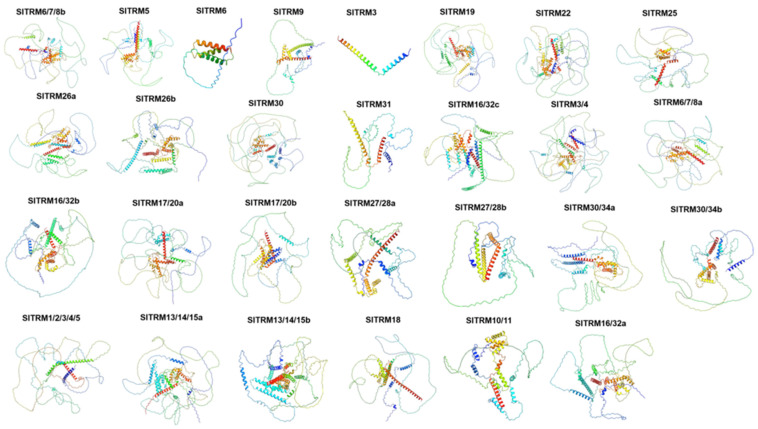
Three-dimensional structure prediction of SlTRM gene family members. Different color depths represent confidence, with increasing confidence from blue to red.

**Figure 5 ijms-26-03676-f005:**
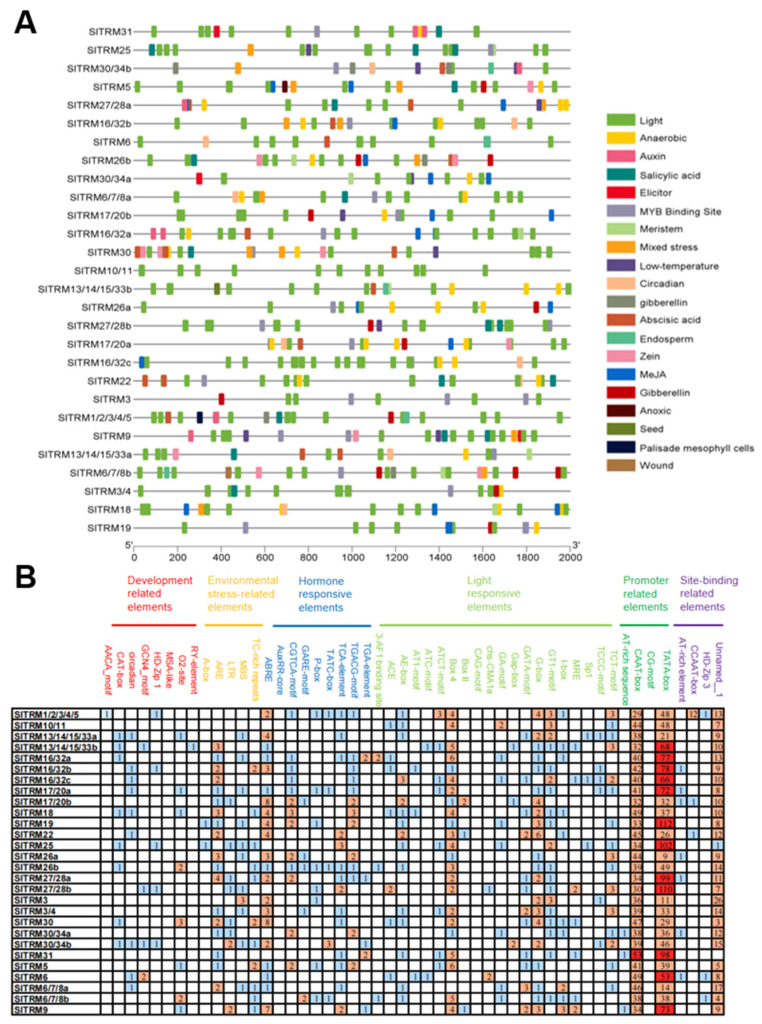
The cis elements in the promoter regions of tomato *SlTRMs*. (**A**) The distribution of cis-acting elements within the gene, with different colors indicating distinct elements. (**B**) Statistical analysis of the cis-acting components. The number in the grid represents the number of elements, and the color from blue to red represents the number of elements from fewer to more.

**Figure 6 ijms-26-03676-f006:**
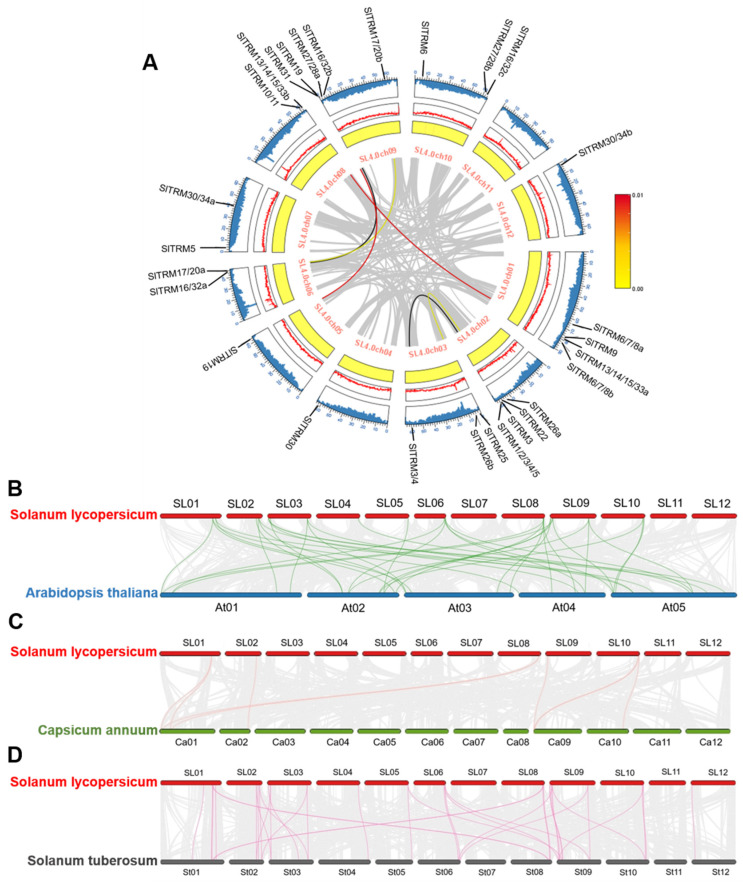
Intra and interspecies collinear analysis of the SlTRM gene family members. (**A**) Advanced circuit diagram of the SlTRM gene family members; gray lines represent all collinear genes in the tomato genome, while red, black, and yellow lines denote the SlTRM genes with collinear relationships. The yellow circle indicates gene density, the red curve illustrates GC content, and the blue curve range represents the N ratio. (**B**–**D**) Interspecies collinear analysis among *Solanum lycopersicum*, *Arabidopsis thaliana*, *Solanum tuberosum*, and *Capsicum annuum*. The gray lines depict the collinear relationships between other genes in the genome, whereas red, green, and pink lines represent the collinear relationships among TRM genes, respectively.

**Figure 7 ijms-26-03676-f007:**
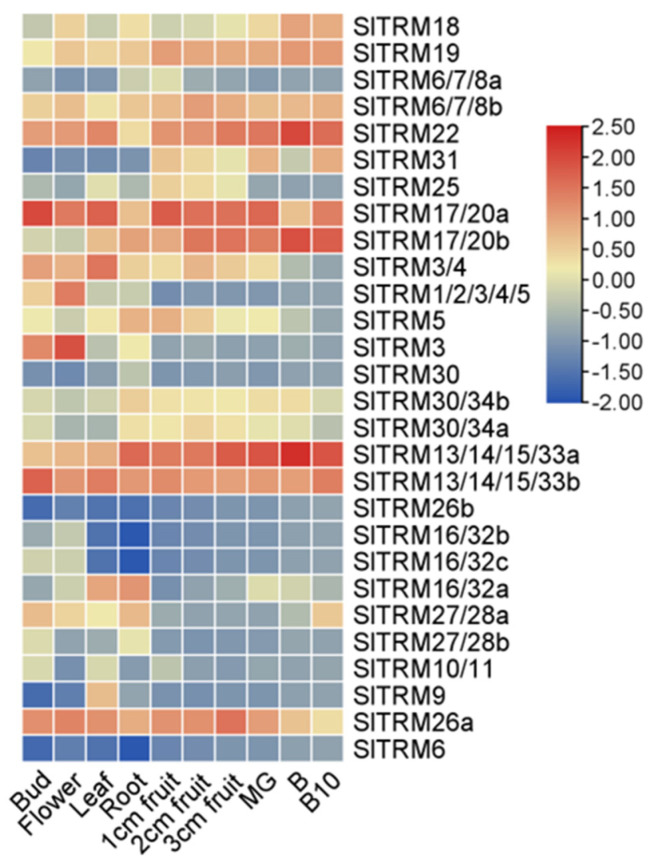
Expression of SlTRM genes in different tissues. Heatmaps were generated following the normalization of absolute fragment count (FPKM) values per kilobase transcript for each gene across various tomato organs. The dendrogram on the left illustrates the results of the intergenic clustering analysis.

**Figure 8 ijms-26-03676-f008:**
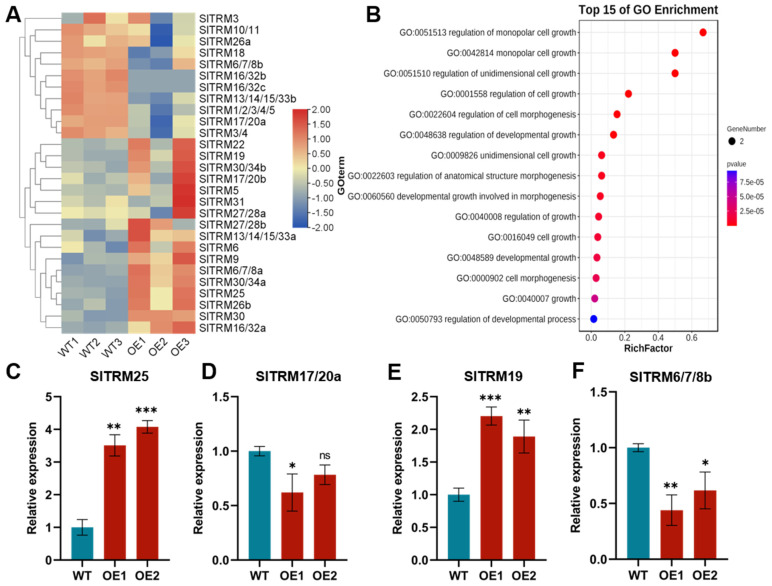
Analysis of the expression and Gene Ontology (GO) enrichment of the SlTRM gene in lateral developmental transcription. (**A**) Expression levels of the SlTRM gene in the lateral developmental transcriptome are presented, where WT denotes wild type and OE indicates overexpressed lines. The color coding is as follows: blue represents low-level expression, while red indicates high-level expression. (**B**) GO enrichment analysis of the SlTRM gene family members is illustrated, with blue and red denoting enriched *p* values. (**C**–**F**) The expression of the tomato SlTRM gene in both wild-type and overexpressed plants was analyzed using qRT-PCR. Asterisks indicate significant differences determined by one-way ANOVA (ns ≥ 0.05, * *p* < 0.05, ** *p* < 0.01, *** *p* < 0.001).

**Figure 9 ijms-26-03676-f009:**
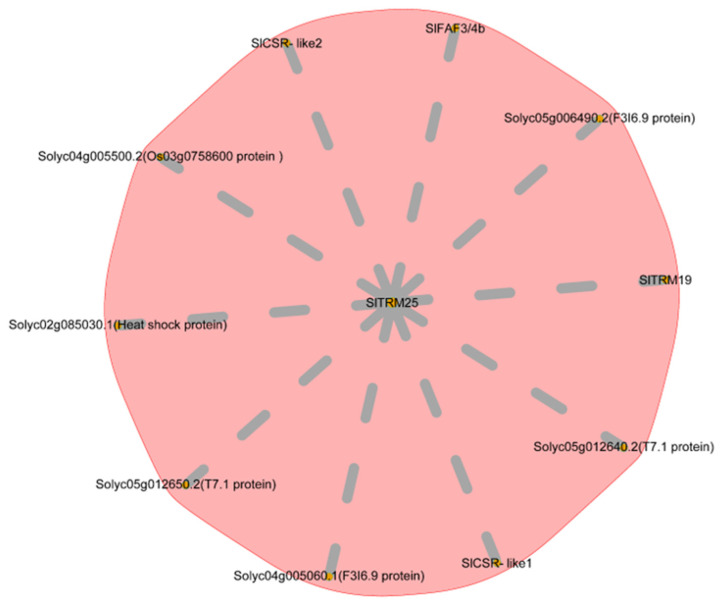
Analysis of the interaction network for the *SlTRM25* gene.

## Data Availability

The data that have been used are confidential.

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
