# Peer review of "Genome-Wide Identification and Expression Analysis of TONNEAU1 Recruited Motif (TRM) Gene Family in Tomato"

_ijms, 2025, doi:10.3390/ijms26083676_

Round 1
Reviewer 1 Report
Comments and Suggestions for Authors
1. Are the expression patterns of key genes consistent with their proposed roles in the biosynthetic pathway?
2. Do the results show any co-regulated gene clusters that suggest coordinated pathway regulation?
3. Are there any discrepancies between transcript abundance in different tissues ? If so, how can these be explained?
4. How do the observed results contribute to understanding regulatory networks (e.g., TFs, miRNAs) involved in the pathway?
5. Have outlier data points been investigated, and could they indicate biological variation rather than technical artifacts?
6. In Figure 7, why does a group of genes showed a significant increase in expression in some tissues but not in other tissues? Could this suggest an alternative regulatory mechanism?
Comments on the Quality of English LanguageMINOR
Author Response
We sincerely thank the editor and all reviewers for their valuable feedback that we have used to improve the quality of our manuscript. The reviewer comments are laid out below in italicized font and specific concerns have been numbered. Our responseis given in normal font and changes/additions to the manuscript are given in the red text.
Reviewer #1:
1 Comment: Are the expression patterns of key genes consistent with their proposed roles in the biosynthetic pathway?
1 Reply:Thank you very much for your insightful comments. We conducted Gene Ontology (GO) enrichment pathway analysis on all genes within the SlTRM gene family. The results indicated that key genes are significantly enriched in pathways associated with cell growth and development, unipolar cell growth, and cell morphogenesis. We hypothesize that these key genes may play a crucial role in regulating fruit shape and lateral branch development. Furthermore, our preliminary data suggest that these key genes exhibit similar expression patterns to numerous genes involved in lateral branch development, thereby reinforcing the potential involvement of the key genes in the regulation of these traits. Your comments have also encouraged us to further investigate the specific roles of SlTRM genes in the regulation of fruit shape and lateral branch development. Currently, we are conducting gene functional validation experiments on the key genes and analyzing the expression patterns of related genes. We hope to have the opportunity to publish these findings in your journal in the future.
2 Comment: Do the results show any co-regulated gene clusters that suggest coordinated pathway regulation?
2 Reply:Thank you very much for your professional comments. Our results indicate that the SlTRM genes belong to the same gene family and possess identical conserved domains, suggesting they may have identical or similar functions. Furthermore, based on the studies conducted by Wu et al. (2018)[1] and Zhang et al. (2023)[2], which are cited in our literature, their articles investigated the synergistic or antagonistic functions among certain SlTRM genes in regulating fruit traits. For instance, they demonstrated that SlTRM3/4 and SlTRM5 have an additive effect on tomato fruit traits, counteracting the effects of the ovate/slofp20 (o/s) mutants. Additionally, the cumulative mutations of SlTRM19 and SlTRM17/20a further enhance the effects induced by the o/s mutants. Simultaneously, we discovered that many SlTRM genes exhibit either identical or opposite expression trends in the transcriptome, suggesting their involvement in regulating certain traits through similar expression patterns. This phenomenon has prompted further contemplation, and we are currently conducting additional analyses and verifications of the expression patterns and pathway networks of these genes. We anticipate elucidating the gene pathways soon and organizing our findings for submission to your esteemed journal.
3 Comment: Are there any discrepancies between transcript abundance in different tissues ? If so, how can these be explained?
3 Reply:Thank you for your insightful comment. Although the genes in the SlTRM gene family possess conserved structural domains, our study reveals significant differences in the transcriptional abundance of various genes across different tissues. For instance, Sun et al. (2023)[3] found that all genes in the PEBP family are associated with flowering; however, their expression levels during the five distinct stages, from flower bud formation to fruit setting, vary considerably. This suggests that these genes may perform different functions at different developmental stages. Additionally, referencing previous studies, we noted that Zhang et al. (2023)[2] detected varying expression trends of several genes in the SlTRM family across different tissues, including the pericarp, seeds, and central column, at various temporal and spatial levels. In summary, different genes exhibit varying transcriptional abundances across tissues, which may indicate their distinct functional sites and roles. Furthermore, we will conduct extensive experimental assays to validate the expression patterns and functional analyses of these genes.
4 Comment: How do the observed results contribute to understanding regulatory networks (e.g., TFs, miRNAs) involved in the pathway?
4 Reply:Thank you very much for your thoughtful comments. We have conducted a comprehensive analysis of the SlTRM gene family utilizing bioinformatics methods; however, our current analysis is primarily based on predictive models. We have explored their functions through Gene Ontology (GO) enrichment analysis, and our preliminary data suggest the involvement of these genes in lateral branch development. Furthermore, based on existing literature, we have identified that genes within this family are linked to the regulation of fruit shape and lateral branch development. Our unpublished data indicate that SlTRM25 may serve as a potential target gene implicated in the regulation of lateral branch development. We are currently performing follow-up validation experiments. Once again, we appreciate your suggestions, and we will thoroughly examine them to further confirm the functions of these genes, with the goal of publishing our research findings in your esteemed journal in the future.
5 Comment: Have outlier data points been investigated, and could they indicate biological variation rather than technical artifacts?
5 Reply:Thank you very much for your detailed comments. In our analysis of this gene family, we employed various bioinformatics approaches to identify the SlTRM genes in tomato. By integrating multiple tools, we conducted a thorough analysis of each characteristic of the SlTRM genes to ensure that all genes met the criteria for classification as SlTRM genes. This included a rigorous examination of gene domains, three-dimensional structures, and fundamental gene information. Furthermore, to analyze the tissue-specific expression of the SlTRM gene family and validate transcriptome and gene expression levels, we performed multiple biological and technical replicates to ensure the reliability of our results and confirm that they were not due to random occurrences. Consequently, the emergence of certain anomalous data leads us to more accurately attribute them to variations during the evolutionary process of genes, a phenomenon that indicates the richness and specificity of their gene family functions. In response to your comments, we will subsequently reanalyze and experimentally validate these genes.
6 Comment: In Figure 7, why does a group of genes showed a significant increase in expression in some tissues but not in other tissues? Could this suggest an alternative regulatory mechanism?
6 Reply:Thank you very much for your insightful comments. We apologize for not clearly addressing this issue in our previous submission. Although members of a gene family share a common origin and exhibit similar functions, their expression patterns are not identical. Consequently, their expression varies across different tissues and developmental stages, which supports your suggestion that there may be alternative regulatory mechanisms among SlTRM genes. We examined the cis-acting elements in the promoter regions of these genes and found that the distribution and types of these elements differ, indicating that they may be regulated by distinct factors under various environmental conditions during growth and development. Furthermore, their differential expression across tissues suggests the possibility of coordinated regulation or spatiotemporal alternative regulation among these genes. In the future, we will conduct a thorough investigation into the spatiotemporal expression of these genes and the regulatory mechanisms between upstream and downstream genes to explore the complex functional interactions among them.

Reviewer 2 Report
Comments and Suggestions for Authors
Please find attachment

Author Response
We sincerely thank the editor and all reviewers for their valuable feedback that we have used to improve the quality of our manuscript. The reviewer comments are laid out below in italicized font and specific concerns have been numbered. Our responseis given in normal font and changes/additions to the manuscript are given in the red text.
Reviewer #1:
1 Comment: The introduction outlines the role of the TRM gene family in plant growth and mentions fruit shape diversity in tomatoes, referencing genes like SUN and OVATE. However, it lacks a clear connection between TRM genes and tomato fruit morphology, as well as a justification for the study’s focus. Enhance the background by explicitly linking TRM genes to tomato fruit shape regulation. For
example, cite prior studies such as Wu et al., 2018, which may provide evidence of TRM involvement in fruit morphology. Include a brief review of existing research on tomato TRM genes to establish the current knowledge gap. This will underscore the novelty and significance of your study, making a compelling case for its contribution to the field.
1 Reply:Thank you for your valuable comments, and we apologize for not adequately addressing this issue in our previous submission. In response, we have incorporated relevant literature regarding the topics you raised, specifically referencing Arabidopsis TRM1 and TRM2, tomato TRM5, as well as rice and wheat GW7. These studies illustrate that TRM genes promote shape elongation by enhancing cell proliferation along the proximal-distal (circumferential) axis while concurrently reducing the number of cells in the medio-lateral direction. Furthermore, our revisions detail how TRM genes regulate cell arrangement through microtubule dynamics, thereby influencing the morphology of fruit organs. We have thoroughly revised the text to include these insights, and we have introduced additional studies on SlTRM genes. These enhancements have significantly improved the cohesiveness of the article. We sincerely appreciate your insightful question.
2 Comment: The classification of tomato and Arabidopsis TRM genes into five phylogenetic subfamilies is reported, but their biological significance is not discussed. Explore the functional implications of these subfamilies. For example, reference known roles of Arabidopsis TRM genes (e.g., AtTRM5 in leaf growth) to hypothesize conserved or divergent functions in tomato.
2 Reply:We greatly appreciate your insightful comments. We have referenced reports from other research scholars, such as the regulation of leaf growth by microtubule dynamics in Arabidopsis AtTRM5, the control of cell division by AtTRM7, and the regulation of flavonoid metabolism by TRM21. Consequently, we hypothesize that the homologous genes in tomatoes may exhibit similar functions. These findings are significant for enhancing our understanding of the roles of other TRM genes in various metabolic activities in tomatoes. We plan to experimentally validate these predicted functions to further clarify the relationships between homologous gene functions. We have incorporated these revisions into the text and thank you once again for your valuable comments.
3 Comment: SITRM3 and SITRM6 are noted to have only one coding sequence (CDS), unlike other SITRM genes, but this observation lacks further exploration. Discuss the potential functional consequences of this structural simplicity. Draw on known TRM functions in tomato or related species to support your interpretation.
3 Reply:We sincerely apologize for the oversight in our response to your comments and for not thoroughly addressing the issues you raised in the article. We have referred to other studies, particularly regarding the tomato TRM family members (SlTRM5, SlTRM19, and SlTRM17/20a), which exhibit complex structures containing multiple coding sequences (CDS) and regulate fruit morphogenesis through microtubule-dependent cytoskeleton reorganization. Our analysis indicates that SlTRM3 and SlTRM6 possess simpler structures, suggesting they may play a more significant role in maintaining fundamental functions such as cytoskeletal integrity. We speculate that these genes may have unique and specialized functions. In future studies, we will conduct expression pattern and functional analyses of these two genes to investigate the underlying reasons for this phenomenon and elucidate their primary functions. We have carefully revised these sections in the manuscript.
4 Comment: AlphaFold3 is used to predict 3D structures of SITRM proteins, but the analysis is superficial. Select representative proteins (e.g., SITRM3, SITRM6, and a more complex protein like SITRM5) and describe specific structural features, such as alpha-helices or beta-sheets.
4 Reply:Thank you very much for your kind comments. Structural characterization has revealed distinct secondary structural features among the SlTRM isoforms. We conducted a further analysis of the characteristics of these genes and found that the typical SlTRM5 exhibits a predominantly disordered conformation (82.75% random coil) with the least α-helical content (16.38%). This is in stark contrast to the highly ordered structures of SlTRM3 (66.20% α-helix; 33.80% random coil) and SlTRM6 (64.60% α-helix; 35.96% random coil). This marked structural dichotomy strongly suggests functional differentiation: the intrinsic disorder of SlTRM5 may facilitate transient protein-protein interactions within the signaling complex, while the stable helical structure of SlTRM3 and SlTRM6 may play a structural role in cytoskeleton organization. This content has been described in detail in the text, and subsequent studies will be conducted on the specific functions of these two genes to explore the functional differences arising from their structures.
5 Comment: Cis-acting elements are identified in SITRM genes, but their relevance to gene
expression is not addressed.
5 Reply:Thank you for your inquiry. We have conducted a thorough analysis of the distribution of cis-elements. Notably, there are variations in the cis-acting elements present in the promoter regions of these genes, which influence their expression under various environmental conditions, such as temperature and light. Given that these regions are critical for gene transcription regulation, we hypothesize that numerous upstream and downstream regulatory genes may be involved. Currently, we are investigating the cis-elements associated with these genes to further validate their distinct expression patterns. Future studies will delve deeper into the expression of these genes under diverse conditions and identify upstream and downstream genes to assess the impact of cis-acting elements. Our goal is to publish our findings in your esteemed journal in the future.
6 Comment: SITRM25 is identified as differentially expressed and potentially linked to lateral branching, but supporting evidence is limited. Propose experimental validation, such as gene knockout or overexpression studies, to confirm SITRM25’s role. Or integrate findings from its interaction network (Section 2.9, Page 12) to bolster the hypothesis, providing a more robust foundation for its functional significance.
6 Reply:Thank you very much for your kind comments. Currently, we have analyzed the expression patterns of the SlTRM25 gene alongside several lateral branch development genes, including BL, BRC1b, and PIN4, and have observed that their expression trends are consistent. Furthermore, based on unpublished data from our research group, we have detected that SlTRM25 may function as a downstream gene involved in lateral branch development. At present, we are conducting functional validation of the SlTRM25 gene by creating transgenic materials to ascertain its role. Additionally, we are analyzing the interaction network of SlTRM25 in the regulation of lateral branches, exploring its relationship with various lateral branch development genes. These findings have been described in detail within the text, and ongoing research is in progress. We hope to publish our research findings in your esteemed journal.

Round 2
Reviewer 2 Report
Comments and Suggestions for Authors
Suggest accepting it in present form